



# Identifying biomass burning impacts on air quality in Southeast Texas 26–29 August 2011 using satellites, models and surface data

David A. Westenbarger[1], Gary A. Morris[2]

[1]Texas Commission on Environmental Quality, Austin, TX, USA
5  [2]School of Natural Sciences, St. Edward's University, Austin, TX, USA

*Correspondence to*: David A. Westenbarger (David.Westenbarger@tceq.texas.gov)



**Abstract.** In this paper, we examine the influence of transported emissions from biomass burning (BB) in regions upwind of the Houston-Galveston-Brazoria (HGB) area on $O_3$ and precursors during the period 26–29 August 2011 using remotely-sensed satellite retrievals from GOES13, MODIS, AIRS and CALIOP. Multiple instruments observed high aerosol optical depth in HGB on 26 August. Surface measurement networks and model estimates (CMAQ, NAAPS) confirm presence of

5   aerosols, smoke and CO along with elevated amounts of $O_3$. Trajectory models (HYSPLIT, BAMGOMAS) trace air masses to possible sources of BB emissions in the Mississippi delta and other upwind regions. Results demonstrate emissions from BB in upwind areas likely contributed to enhanced $O_3$ and precursor concentrations measured by surface monitors in HGB which were characterized by enhancements of 63–71 ppb in daily maximum 1-hr $O_3$ at the Houston East monitor on 26Aug compared to 21–25Aug. This study demonstrates an approach to identifying biomass burning influences on high ozone

10  events which may be useful in determining compliance with EPA NAAQS.



## 1 Introduction

Previous studies have examined transport of O$_3$ and O$_3$ precursors from biomass burning (BB) to downwind regions. Under proper conditions, this transport can contribute to enhancements of surface O$_3$ and O$_3$ precursors. On 26 August 2011 and 29 August 2011, multiple surface monitors in the Houston-Galveston-Brazoria (HGB) area of Texas (TX) observed 1-hr

average O$_3$ concentrations in excess of 120 parts per billion (ppb). Though in and of themselves no isolated events such as these directly result in violations of the Federal National Ambient Air Quality Standards (NAAQS) for O$_3$, when averaged with the fourth highest measurements from other days within a three year period, they could contribute to a violation of the standard.

    In this paper we present evidence linking surface O$_3$ enhancements in HGB on 26Aug and 29Aug to transported

emissions from BB using (a) surface measurements of O$_3$, CO and particulate matter (specifically, PM$_{2.5}$), (b) remotely-sensed satellite retrievals of aerosols and CO, (c) surface retrievals of aerosols, (d) models of atmospheric transport of aerosols and smoke and (e) trajectory models. Though HGB is well known for its O$_3$ challenges, this evidence clearly demonstrates that O$_3$ events on 26Aug and 29Aug were unusual even for this area. The combination of the statistical analysis of the surface monitor data in conjunction with the tracking of smoke plumes in the satellite data proves to be a powerful tool

for investigating high ozone events attributable to biomass burning influences. This approach could well be adapted for application to other pollution events in the HGB area as well as in other regions and at other times.

## 2 Background

As the fourth most populous metropolitan area in the U.S., HGB is home to one of the largest concentrations of petrochemical manufacturing facilities in the world. The area is also home to numerous other industrial facilities, one of the

nation's busiest sea ports, electric generating units, over 2.1 million people and businesses and their associated motor vehicles. Houston's location near the Gulf of Mexico and Galveston Bay and related summertime meteorology all play key roles in development of high levels of O$_3$ often observed in HGB [e.g., Banta et al., 2005].

    Air quality in HGB continues to violate O$_3$ NAAQS though with less frequency and severity than in the past. While the primary contributor to O$_3$ production is local emissions and photochemistry combined with conducive meteorology (flow

reversal, poor ventilation), emissions from upwind BB also can significantly contribute. If transported emissions are contributing in sufficient quantities to violations of O$_3$ NAAQS, local emissions controls alone may not be adequate to attain Federal standards. Assessment of conditions under which transported emissions are contributing to O$_3$ enhancements and the magnitudes of those contributions is critical to a comprehensive understanding of air quality in HGB. As local emissions decline, impacts of transported O$_3$ and its precursors become more important.

An extensive literature links elevated O$_3$ and O$_3$ precursor concentrations with emissions from upwind fires (e.g., Jaffe et al., 2004; Junquera et al., 2005; Lapina et al., 2006; McKeen et al., 2002; Morris et al., 2006; Oltmans et al., 2010; Pierce et al., 2009; Real et al., 2007; Val Martin et al., 2006; Wigder et al., 2013, Forster et al., 2001; Lamarque et al., 2003;





Colarco et al., 2004). Transport of BB emissions can affect $O_3$ concentrations in the troposphere on regional and global scales (Fishman et al., 1992; Ansmann, et al., 2009; Bertschi, et al., 2004). Numerous studies have tracked BB emissions using satellites and models (Cahoon, et al., 1994; Alvarado et al., 2010; Luo et al., 2010; Morris et al., 2006; Fiore, et al., 2014). The U.S. EPA provides a rigorous process through which air control agencies may demonstrate that incidences of

elevated $O_3$ are the result of emissions from upwind fires, are thus "exceptional," and therefore are not preventable with local controls. While the approach presented here may be relevant for investigating exceptional events, it does not entirely satisfy EPA requirements for demonstrating exceptional events but could be an adjunct to them.

## 3 Observations for this High Ozone Event

### 3.1 Surface Ozone

Daily maximum 1-hr $O_3$ at Houston East reached 78.7 ppb 20Aug, beginning a 5-day period (20-24Aug) that never exceeded 66.7 ppb on subsequent days. Overnight lows fell below 2 ppb each night. Beginning 26Aug, daily maximum 1-hr $O_3$ rose to 128.5 ppb and remained elevated (> 90 ppb) for the next four days (26–30Aug). Figure 1 shows the locations of monitors in the HGB and Beaumont/Port Arthur (BPA) regions as well as the maximum 1-hour $O_3$ readings from each monitor on 26 and 29 August 2011. Overnight lows on 26Aug exceeded 20 ppb. From 31Aug-4Sep, daily maximum 1-hr $O_3$ at Houston

East was much lower, ranging from 38.4 ppb to 76.3 ppb.

Panel (a) of Fig. 2 plots daily maximum 1-hr $O_3$ at the Houston East surface Continuous Air Monitoring Station (CAMS) alongside average daily maxima of all surface monitors in HGB, all CAMS in the BPA area in southeast TX, and all of the CAMS in Louisiana (LA) and Mississippi (MS) combined. All regions observed similar patterns with daily maximum 1-hr $O_3$ being relatively low in the first period (20–24Aug), elevated in the second (26–30Aug) and lower again in

the third (31Aug–4Sep), though the differences between the first and second periods is smallest in the LA/MS data nearest the source of the biomass burning (as shown below).

Histograms of daily maximum 1-hr average surface $O_3$ concentrations are plotted in Fig. 3 comparing $O_3$ concentrations across 36 surface monitors in HGB (panel a-1) and 9 in BPA (panel a-2) across three periods: (1) before arrival of BB emissions (20–24Aug), (2) during the multi-day smoke event in HGB (26–30Aug) and (3) after the smoke event began to

dissipate (31Aug–4Sep). 25Aug was excluded because of precipitation on that day. Most values in the first period in HGB were low (65 % < 60 ppb); only 13 % exceeded 75 ppb. The second period saw a dramatic shift to a higher ozone regime across the entire HGB area with no daily 1-hr maxima below 60 ppb and 89 % above 75 ppb. The third period saw a return to a regime similar to the first with 74 % of daily maxima below 60 ppb and only 5 % above 75 ppb. This marked shift in the entire distribution suggests that the regime that arrived 25–26Aug brought BB emissions that contributed to ozone formation

across the entire HGB area.

Patterns similar to the pattern seen in HGB are evident at monitors in BPA and upwind states. The distribution of daily maximum 1-hr average surface $O_3$ concentrations in BPA was skewed left in the first period (86 % < 60 ppb) and right in the





second (83 % > 75 ppb). The third period saw a return to the left skew pattern seen in the first period (91 % < 60 ppb). This result indicates that BPA was experiencing a similar phenomenon to the one that influenced HGB; the $O_3$ pattern observed across HGB was not an isolated or purely local phenomenon, suggesting a larger scale influence than simply locally generated pollution in HGB.

## 3.2 Meteorological Observations

The HGB area on the Gulf of Mexico coast of southeast TX, USA, is known to experience complex meteorological phenomena that contribute to its ongoing though improving air quality challenges. These conditions include recirculation due to marine flow reversals (e.g., Banta et al., 2005), long-range transport (e.g., Morris et al. 2006), and stratosphere-troposphere exchanges (e.g., Neu et al., 2014). During August 2011 HGB experienced many hot (> 37° C) days except for 25Aug when a cold front approaching from the northeast became stationary over southeast TX and the lower delta states. This system reduced temperatures considerably across the region. Further, during the final week of the month, the area experienced mostly dry conditions with light winds and deep boundary layer heights (~2-3 km) based on meteorological measurements at Houston Hobby Airport (KHOU). This boundary resulted in precipitation across HGB on 25Aug, and the meteorology was conducive to bringing air into HGB from LA and MS.

Wind speeds were moderate and steady (0.007–8.0 m s$^{-1}$, mean 2.1 m s$^{-1}$) from 25 Aug through 31Aug but increased dramatically 1–4Sep (0.02–16.9 m s$^{-1}$, mean 4.4 m s$^{-1}$). Wind directions were consistently from the east-southeast through west-southwest over the period 19–24Aug, rotating in a "flow reversal" pattern. This pattern is typical of the area and has been shown to contribute to surface ozone enhancements through recirculation of the Houston urban and industrial plumes (e.g. Banta, et al., 2005). Beginning 25Aug wind directions became more variable, arriving at the monitor from the north-northeast early in the morning, rotating throughout the day, and remaining variable through 29Aug.

## 3.3 Satellite Fire Detection

During the week preceding and including the elevated $O_3$ concentrations measured in HGB on 26Aug and 29Aug, several satellite-based fire detection systems (e.g., NASA's Fire Information for Resource Management System (FIRMS), NOAA's Hazard Mapping System (HMS), MODIS Active Fire and Burned Area) detected a large number of fires burning throughout the continental U.S. Of particular relevance to this study were fires burning in the Mississippi delta states of LA and MS. Supplementary material presents details of these detection systems. Figure 4 presents a representative map of suspected fire locations detected by MODIS during 24–26Aug. Table SM-1 provides details of satellites used in this analysis.

## 4 Supporting Evidence for Biomass Burning Influences

This section presents an approach to evaluating the influence of biomass burning on the high ozone event of 26–29 August 2011. In particular, we examine additional surface observations of PM$_{2.5}$, CO and aerosol optical depth (AOD); satellite





observations of AOD, aerosol type, and CO; and model outputs of air mass trajectories and aerosol distributions. Our analysis demonstrates the influence of biomass burning on the surface measurements along the Gulf Coast and connects the perturbed, high-ozone air mass over HGB and southeast TX to upwind fire events in LA and MS. This approach can be applied to other cases for which biomass burning events could contribute to higher ozone concentrations.

## 4.1 Surface Monitors

### 4.1.1 Particulate matter and aerosols

Daily maxima and histograms of the distributions of $PM_{2.5}$ concentrations measured at surface monitors show patterns similar to $O_3$ with lower values at the beginning of the period, higher values in the middle and a return to lower values at the end. Daily maximum 1-hr $PM_{2.5}$ at Houston East ranged from 12.2 to 22.6 µg m$^{-3}$ over 20–24Aug, rose to a range of 24.7– 39.6 µg m$^{-3}$ over 26–30Aug, then ranged from 10.6 µg m$^{-3}$ to 33.6 µg m$^{-3}$ over 31Aug–4Sep. Panel (b) of Fig. 2 plots daily maximum $PM_{2.5}$ at Houston East alongside average daily maxima of surface monitors in HGB, BPA and LA and MS over the three periods detailed above (20–24Aug; 26–30Aug; 31Aug–4Sep). Patterns similar to Houston East were observed at HGB and BPA monitors.

Histograms of daily maximum $PM_{2.5}$ plotted in Fig. 3 for 12 HGB (panel b-1) and 4 BPA (panel b-2) monitors show similar patterns to those seen above for $O_3$: distributions with many lower values in the first period (HGB: 99 % < 12 µg m$^{-3}$; BPA: 100 % < 12 µg m$^{-3}$), a shift to much higher values in the second period (HGB: 100 % > 12 µg m$^{-3}$; BPA: 100 % > 12 µg m$^{-3}$) and a return to lower values in the third (HGB: 33 % < 12 µg m$^{-3}$, BPA: 33 % < 12 µg m$^{-3}$).

### 4.1.2 AERONET aerosols

Panel (c) of Fig. 2 plots daily maximum AOD by mode retrieved by the AERONET instrument atop the Moody Tower at the University of Houston for 20Aug through 3Sep (no data was available for 4Sep). Note that both peak total AOD and peak fine mode AOD during this period occurred on 26Aug with a lower peak on 31Aug. Fine mode aerosols are those most associated with BB. Histograms of daytime hourly AERONET total column AOD (unitless) and fine mode AOD (unitless) from the University of Houston (UH) site for the three periods are plotted in Fig. 3 panels (c-1) and (c-2), respectively. These plots once again demonstrate the pattern observed previously in other measures: the distribution during the first period exhibited mostly lower values (88 % < 0.3). The second period saw an increase of higher values (49 % > 0.3). The third period saw a return to lower values and few higher values (82 % < 0.3). Fig. 3 panel (c-2) shows the pattern was even more pronounced for fine mode AOD (first period: 100 % < 0.3; second period: 34 % > 0.3; third period: 95 % < 0.3). This evidence suggests that the system that arrived in HGB 25–26Aug brought a significant loading of fine mode aerosols.

Figure 3 panel (c-3) plots daily maximum fine mode fraction AOD from the UH AERONET site over the period 20Aug–3Sep. This plot mimics Fig. 2 panel (d) with local peaks on 26Aug and 31Aug suggesting AOD increases on those days were driven largely by fine mode aerosols. Daily maximum fine mode fraction ranged in a narrow band of 0.79–0.86





from 20-24Aug (mean 0.82), rose to 0.96 on 26Aug and remained in a range of 0.80–0.94 (mean 0.89) over 27–31Aug, then fell to a range of 0.56–0.81 over 1–3Sep (mean 0.67). AERONET retrievals on 31Aug appear to have been influenced by remnants of BB emissions and so it was classified into the middle period for this analysis.

Histograms of the AERONET fine mode fraction in Fig. 3 panel (c-3) repeat the pattern seen above: higher frequencies of lower values in the first period (20–24Aug: 79 % of observations < 0.8, a shift to higher values in the second period (26–30Aug: 46 % > 0.8), and subsequent return to higher frequencies of lower values in the third period (31Aug–3Sep: 82 % < 0.8).

### 4.1.3 CO

Panel (e) of Fig. 2 plots daily maximum 1-hr average CO concentrations (ppm) at the Clinton (CAMS 403) surface monitor and mean daily maxima at all monitors in HGB, BPA and LA and MS from 20Aug–4Sep. The Clinton monitor was chosen because it is the closest monitor to Houston East that has a CO instrument (~5.25 km). When examined in conjunction with $O_3$ (panel a) and $PM_{2.5}$ (b), we see replicated the same pattern of lower daily maximum CO concentrations before and after the BB event in HGB and higher values during the multi-day event.

Histograms of daily maximum 1-hr average CO concentrations plotted in Fig. 3 panel (d-1) repeat the pattern: higher frequencies of lower concentrations (54 % < 0.20 ppb) and fewer higher concentrations (17 % > 0.25 ppb) in the first period (20–24Aug), a shift to lower frequencies of lower concentrations (33 % < 0.20 ppb) and many more higher concentrations (47 % > 0.25 ppb) in the second period (26–30Aug), followed by a return to higher frequencies of lower concentrations (63 % < 0.20 ppb) and lower frequencies of higher concentrations (30 % > 0.30 ppb) in the third period. The shift to higher values in the second period is remarkable for the large fraction > 0.30 (37 %) compared to either other period (7 % and 13 %). This result demonstrates that CO, a tracer of BB emissions, followed patterns similar to $O_3$ and $PM_{2.5}$ across the entire HGB region over the period. This pattern was echoed in BPA.

Figure 3 repeats the daily maximum 1-hr average CO histogram for the two surface monitors in BPA (panel d-2). The pattern of surface CO in BPA is even more striking than in HGB: pervasive low CO concentrations (100 % < 0.20 ppm) in the first period followed by a shift to higher values in the second (70 % > 0.25 ppm; 50 % > 0.30 ppm) and a return to lower values in the third (50 % < 0.20 ppm; 10 % > 0.30 ppm). This result supports findings for both $O_3$ and $PM_{2.5}$ in HGB and BPA that these compounds were detected at the surface in higher concentrations during the period influenced by BB emissions (26–30Aug) than either before or after.

### 4.2 Satellites

### 4.2.1 AIRS

Figure 5 shows CO VMR (volume mixing ratio) created with the Giovanni visualization tool from daytime (ascending) retrievals by the AIRS instrument on the Aqua satellite for each day of 24–26Aug for each of the three lowest atmospheric





pressure levels reported: 850 hPa, 925 hPa and 1,000 hPa. These retrievals show CO plumes in LA and eastern TX on 24Aug that are cleared away by arrival of the cold front on 25Aug. Fresh plumes of CO arrive over the HGB area, LA and east TX again on 26Aug at all three of these pressure levels.

AIRS total column CO retrievals provide a more robust view of CO than the profiles, though the number of valid
retrievals during the time period of interest is low (92 in HGB, 27 in BPA, zero on several days). Figure 2 panel (f) plots daily maximum AIRS total column CO over 21Aug–4Sep (no valid retrievals on 20Aug, 27Aug, or 3Sep). Here again we see the pattern of lower values early in the period, higher values in the middle period and a return to lower values in the final period. Daily maximum total column CO peaked 25Aug then dropped, peaking again on 31Aug. Histograms of AIRS total column CO plotted in Fig. 3 panels (e-1) and (e-2) recapitulate the pattern seen throughout this paper with low frequencies of
high values before and after the multi-day BB emissions event in HGB and BPA and high frequencies during.

**4.2.2 MODIS & GOES**

Figure 6 panel (a) maps AOD retrievals from the MODIS (MODerate resolution Imaging Spectroradiometer) instrument (3-km resolution product) on board the Terra satellite for overpasses of the southeastern U.S. on 26Aug. Retrievals from MODIS instruments aboard both Terra and Aqua platforms were evaluated. A large plume of AOD to the northeast of the
study area on 24Aug (not shown) was transported southwest over southern MS and LA on 25Aug. The plume was further transported over southeast TX and HGB on 26Aug and is visible in this plot. AOD ranging from 0.00 to 0.33 over 21–25Aug increased to 0.31 (Aqua) and 0.39 (Terra) on 26Aug, peaking for the 19–31Aug period at 0.41 and 0.45 on 27Aug.

Figure 6 panel (b) maps the portion of over-land aerosol retrievals that are fine mode from the MODIS instrument on board Terra for 24–26Aug. Mean fine mode aerosol fractions in MS ranged from 0.80 to 0.99 over the 20–27Aug period,
dropping to 0.16 on 28Aug. In southern LA mean fine mode aerosol fractions ranged from 0.38 to 0.63 over the 20–24Aug period, increasing to 0.70 to 0.74 over the 25–27Aug period before dropping to 0.04 on 28Aug. Southeast TX followed a similar pattern with mean fine mode aerosols ranging from 0.13 to 0.25 over 20–24Aug, rising to 0.68 to 0.88 over the 25–27Aug period, dropping to 0.05 to 0.18 over 28–29Aug and rising again to 0.69 on 30Aug. In HGB, mean fine mode aerosol fractions ranged from 0.23 to 0.57 from 20–25Aug, rising to 0.82 on 26Aug, 0.87 on 27Aug and 0.89 on 30Aug. These
results indicate that large plumes dominated by fine mode aerosols, those most associated with BB, were present over HGB during the multi-day $O_3$ event.

Figure 2 panel (g) plots daily mean AOD from MODIS level 2 3-km retrievals (Aqua and Terra) for HGB and upwind regions for 20Aug–4Sep. These plots quantify patterns observed in mapped AOD retrievals in Fig. 6 panel (a) and confirm GOES East retrievals in Fig. 6, panel (c). AOD was elevated and variable in the upwind areas over the week prior to 26Aug
while HGB exhibited comparatively low AOD over the prior week before an upward trend began 24Aug which persisted through 27Aug as the BB plume arrived.

The GOES (Geostationary Operational Environmental Satellite) system maintained by NOAA consists of weather satellites in geosynchronous orbit with the Earth that also provide retrievals of aerosols. GOES13 over the eastern U.S.

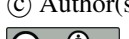



provides retrievals at 15 and 45 minutes past the hour during daylight hours. See Supplementary material for details on GOES. Figure 6 panel (c) maps GOES13 AOD from the retrieval at 19:15 UTC on 26Aug that corroborates AOD retrieved by MODIS above.

### 4.2.3 CALIOP

The CALIOP instrument on the CALIPSO platform provides a truly unique perspective on aerosols in the atmosphere. Though its swath width is quite narrow (80 m), a CALIOP "curtain" provides vertical resolution of 30 m from the surface up to 8 km altitude. Horizontal resolution along-track is ~1/3 km. Though the instrument does not quantify the amount of aerosol present, interpretation of the sizes and shapes of particles using retrieved attenuated backscatter and polarization provide sufficient information to determine the types of features present in a column.

Figure 8 presents aerosol subtypes from 54 vertical feature masks of 5 km along-track CALIOP curtains from the surface to 8 km altitude from the orbital path that traversed HGB on 26Aug. CALIOP algorithms classify aerosol features into categories including: dust, smoke, polluted air, clean continental, polluted continental, clean marine and clear air. White pixels indicate invalid or missing data. CALIOP detected the presence of a ~0.75 km thick layer of "polluted air" at the surface extending roughly 158 km from a point over the Gulf of Mexico to a point about 21 km south-southeast of
Galveston, TX (point a). Above this layer CALIOP detected "clear air" with no detectable aerosols up to 8 km altitude. Beginning in the next curtain one 5-km position further along the orbital path (point b) and extending ~75-80 km inland to a point just north of Interstate 10, CALIOP detected a ~0.5 km thick layer of smoke from ~3.75 km to ~4.25 km altitude. This layer overlaid a layer of "clean marine" from the surface to ~3.1 km altitude which is consistent with the typical onshore flow from the Gulf seen in this and other coastal regions. Above these layers, CALIOP detected "clear air." Beyond this
point (c), CALIOP detected a thick layer of "smoke" from the surface to ~2.6 km altitude, extending ~155–160 km to a point southeast of Crockett, TX near Davy Crockett National Forest. CALIOP detected clear air above this layer of smoke. At point (d) the profile changed to "polluted continental" in the lowest ~3 km with "clear air" above. This evidence indicates that by 26Aug smoke had arrived in HGB, filling the boundary layer, and that by the time of the early afternoon CALIPSO overpass a clear gradient had developed between marine air south of I-10 and smoke-filled air to the north. This observation
complements findings for AIRS CO above that showed plumes moving onshore in the vicinity of HGB near Galveston Bay on 26Aug after traversing the Gulf Coast westward from LA.

### 4.3 Model Data

### 4.3.1 Aerosol Transport Model

The Naval Research Laboratory (NRL) maintains the Naval Aerosol Analysis and Prediction System (NAAPS) model of
aerosols that predicts presence and transport of smoke and other aerosols across the globe. Figure 7 presents contour maps of NAAPS predictions of surface layer smoke concentrations ($\mu$g m$^{-3}$) over North America at 18:00Z for 24–26Aug (see





Supplementary material for details of the NAAPS model). NAAPS predicted a large area of smoke over the continental U.S. on these three days stretching east from the Pacific northwest through the Great Plains over portions of the Midwest, the eastern seaboard, into TX and the southeast U.S. Large non-contiguous plumes, some containing elevated concentrations of smoke, were predicted throughout the U.S., southeast TX and MS delta region in all model runs from 20Aug through 3Sep.

Figure 2 plots average daily NAAPS estimated smoke concentrations ($\mu g\ m^{-3}$) for the 5 model layers closest to the surface in the four $1° \times 1°$ model grid cells encompassing HGB (29.5° N-30.5° N, 94.5° W-95.5° W) (panel h) and the two $1° \times 1°$ model grid cells encompassing BPA (29.5° N-30.5° N, 93.5° W) (panel i). The grid cells in panel (h) contain all of Houston inside Beltway 8, most of Harris, Brazoria and Fort Bend counties, and ~34 surface $O_3$ monitors. Daily averages are computed from 4 daily predictions made at 6-hr intervals (0:00, 6:00, 12:00 and 18:00 UTC). The figures trace a pattern

observed over multiple instruments, models and parameters: lower values during the period before the occurrence of high $O_3$, roughly 20–24Aug, higher values beginning ~26Aug after passage of a cold front on 25Aug that brought BB emissions to HGB and continuing for about 5 days through ~30Aug, followed by return to lower values beginning ~31Aug through 4Sep. The similarity between HGB and BPA indicates that smoke from BB emissions was also affecting the BPA area between HGB and LA to the east over roughly the same period.

The now familiar pattern is also evident in histograms of NAAPS model smoke predictions plotted in Fig. 3 comparing smoke across the entire HGB (panel f-1) and BPA (panel f-2) regions over the three periods. The HGB distribution (panel f-1) is skewed to the left in the first period as most values were low (86 % < 8 $\mu g\ m^{-3}$) with a few pockets of higher concentrations. The second period showed a dramatic shift to higher values across the entire region (60 % > 8 $\mu g\ m^{-3}$). By the third period, smoke across HGB had begun returning to pre-event conditions (61% < 8 $\mu g\ m^{-3}$). This shift in the entire

distribution is consistent with arrival of BB emissions that affected the entire HGB area. The situation in BPA (panel f-2) was nearly identical before (100 % < 8 $\mu g\ m^{-3}$), during (35 % > 8 $\mu g\ m^{-3}$), and after (55 % < 8 $\mu g\ m^{-3}$) the BB event.

### 4.3.2 Backward Trajectories

NOAA's Hybrid Single Particle Lagrangian Integrated Trajectory (HYSPLIT) model [Stein et al., 2015; Rolph, 2017] was used to compute backward trajectories terminating at selected altitudes above Houston East at 19:00 UTC on 26Aug, the

time of the first elevated $O_3$ event (see Supplementary material for details of inputs to the HYSPLIT model). Trajectories terminating at lower altitudes (250 m or below) tended to originate from the northeastern Gulf of Mexico, southeastern LA and southern MS, the latter two being areas of concentrated fire activity during the days before the 26Aug ozone event. These trajectories travelled south over land then west over the Gulf of Mexico near the coast before coming ashore west of Port Arthur, TX, in the vicinity of Galveston Bay on 25Aug (Fig. 4). From there, these trajectories moved west into HGB,

circled clockwise around the city and arrived at the monitor from the east on 26Aug. Most of these trajectories remained within the mixing layer for the majority of their journeys. The residence time in the HGB area suggests that the transported BB plumes interacted with the local plume, intensifying $O_3$ production as a result, similar to what was observed in the Alaskan fire event of 2004 [Morris et al., 2006].



Trajectories that terminated at altitudes from 750 m up to 2.25 km at Houston East travelled west over southern MS and central LA where they encountered intense fire activity three to five days prior to the ozone event on 26Aug. These trajectories then traversed southeast TX before arriving in HGB, occasionally traveling within the mixing layer, but mostly above it. Trajectories terminating from 2.5–5.0 km altitude at Houston East travelled south over AR and MS, then southwest over LA and southeast TX before arriving at the monitor on 26Aug. Trajectories at all of these levels experienced strong descent overnight on 25Aug and into the morning of 26Aug. This strong downward motion suggests air parcels containing emissions from BB in upwind areas could have been brought down into the mixing layer and even to the surface on 26Aug. Kinematic backward trajectory analysis using NASA BAMGOMAS available from the AERONET Web site (aeronet.gsfc.nasa.gov) confirmed HYSPLIT results.

## 5 Analysis

It is important to note the timing of the enhancements of $O_3$, CO, and $PM_{2.5}$ observed in each of the regions of the analysis: HGB, BPA, and LA/MS. In general, we see coherence in the response of all of these indicators, suggestive of the BB influence. Furthermore, the multi-day histograms clearly show that the period of 26–21Aug was perturbed relative to the periods before and after. The NAAPS model result seems to suggest the strongest influences in early September after the surface data indicate that the event had concluded. This result suggests perhaps some difficulties in the model simulation with respect to transport.

## 6 Conclusion

In this paper, we examined the influence of transported emissions from BB on $O_3$ levels in southeast TX on 26Aug and 29Aug using remotely-sensed satellite retrievals of aerosols and/or CO from GOES13 (GOES East), MODIS (Terra and Aqua), AIRS (Aura) and CALIOP (CALIPSO). We compared these with in situ measurements from ground based networks and model estimates (CMAQ, NAAPS). We also used HYSPLIT and BAMGOMAS to trace backward trajectories from HGB to identify possible sources of BB emissions. Both trajectory models revealed strong descent in the hours prior to the high ozone event at the Houston East monitor. These events were characterized by enhancements of 63–71 ppb in daily maximum 1-hr $O_3$ at the Houston East monitor on 26Aug compared to average daily maxima over 21–25Aug.

Both GOES and MODIS identified areas of dense aerosols upwind of HGB in the days preceding the event on 26Aug and increasing aerosol concentrations in southeast TX on 26Aug and 29Aug. CALIOP profiles detected smoke in the lowest 2.5 km of the troposphere from Galveston Bay north to I-10 in a path along the eastern edge of HGB. Two models (CMAQ, NAAPS, respectively) identified particulate matter and smoke in HGB on 26Aug11 and in upwind regions in the preceding days. Trajectory models (HYSPLIT, BAMGOMAS) confirmed that parcels arriving in HGB on 26Aug originated in areas experiencing substantial BB in the preceding days. This combination of evidence suggests that elevated concentrations of $O_3$



measured at surface monitors in HGB and southeast TX on 26Aug and again on 29Aug were likely influenced by emissions from BB in upwind regions such as the delta states of AR, LA and MS as well as potentially western regions of the U.S. Emissions from the BB regions moved westward along the Gulf Coast and then northward into the HGB area, influencing observations of CO, $PM_{2.5}$, AOD, and ultimately, $O_3$.

We outlined an approach that uses histograms of surface data prior to, during, and subsequent to the perturbed period to highlight the unusual influences from the BB. When combined with the satellite data tracking the smoke plume, the trajectories, and the model results, we demonstrate a strong case for BB influences on surface $O_3$ concentrations. This approach can be applied to other locations and/or times to identify exceptional events involving long-range transport of chemical species that negatively impact air quality downwind.

**Competing interest:** The authors declare that they have no conflicts of interest.

**Disclaimer**: N/A

**Special issue statement**: N/A

**Data availability**: All data used here is publicly available from the sources cited.

**Acknowledgements**. The authors thank Yunsoo Choi of the University of Houston and Jim Smith, Mark Estes and Erik
Gribbin of the TCEQ for their insightful comments and suggestions as well as TCEQ Monitoring Division for maintaining the network of surface monitors. We also thank the University of Houston, Barry Lefer, James H. Flynn and their staff for their efforts in establishing and maintaining the University of Houston AERONET site and several HGB area surface monitors, and also Rudy Husar for maintaining the datafed.net Web site. Surface measurements are distributed by the U.S. EPA (www.epa.gov/airdata) in partnership with the TCEQ (www.tceq.state.tx.us/), Louisiana Dept. of Environ. Quality
(www.deq.louisiana.gov/portal) and Mississippi Dept. of Environ. Quality (deq.state.ms.us). AIRS data are distributed by NASA Goddard Earth Science Data Information and Services Center (GES DISC): airs.jpl.nasa.gov/data/get_data. Accessed: 21Apr2015. GOES data are distributed by NOAA National Centers for Environmental Information (NCEI): www.ncdc.noaa.gov/data-access. Accessed 29Aug2014. MODIS data are distributed by NASA Goddard Space Flight Center (GSFC): ladsweb.nascom.nasa.gov/data/. Accessed 20May2015. MODIS Fire data are distributed by NASA Earth
Observing System Data and Information System (EOSDIS) and datafed.net: webapps.datafed.net/Core.uFIND. Accessed: 4Sep2014. Some analyses and visualizations used in this paper were produced with the Giovanni online data system developed and maintained by NASA GES DISC. AERONET data are distributed by NASA GSFC: aeronet.gsfc.nasa.gov/new_web/index.html. Accessed 21Nov2014. NAAPS data are distributed by the Naval Research Laboratory and datafed.net: webapps.datafed.net/Core.uFIND. Accessed: 29Jul2013. HYSPLIT data are distributed by
NOAA Air Resources Laboratory: ready.arl.noaa.gov/HYSPLIT.php. Accessed: 17Jul2014.





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



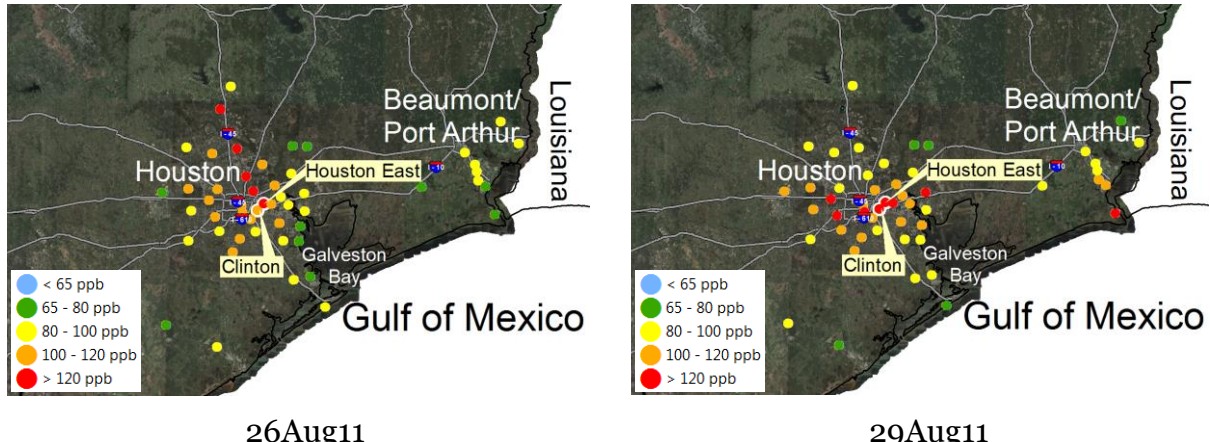

26Aug11     29Aug11

**Figure 1:** Daily maximum 1-hr O$_3$ concentrations at surface monitors in HGB and BPA. Left: 26Aug11. Right: 29Aug11.

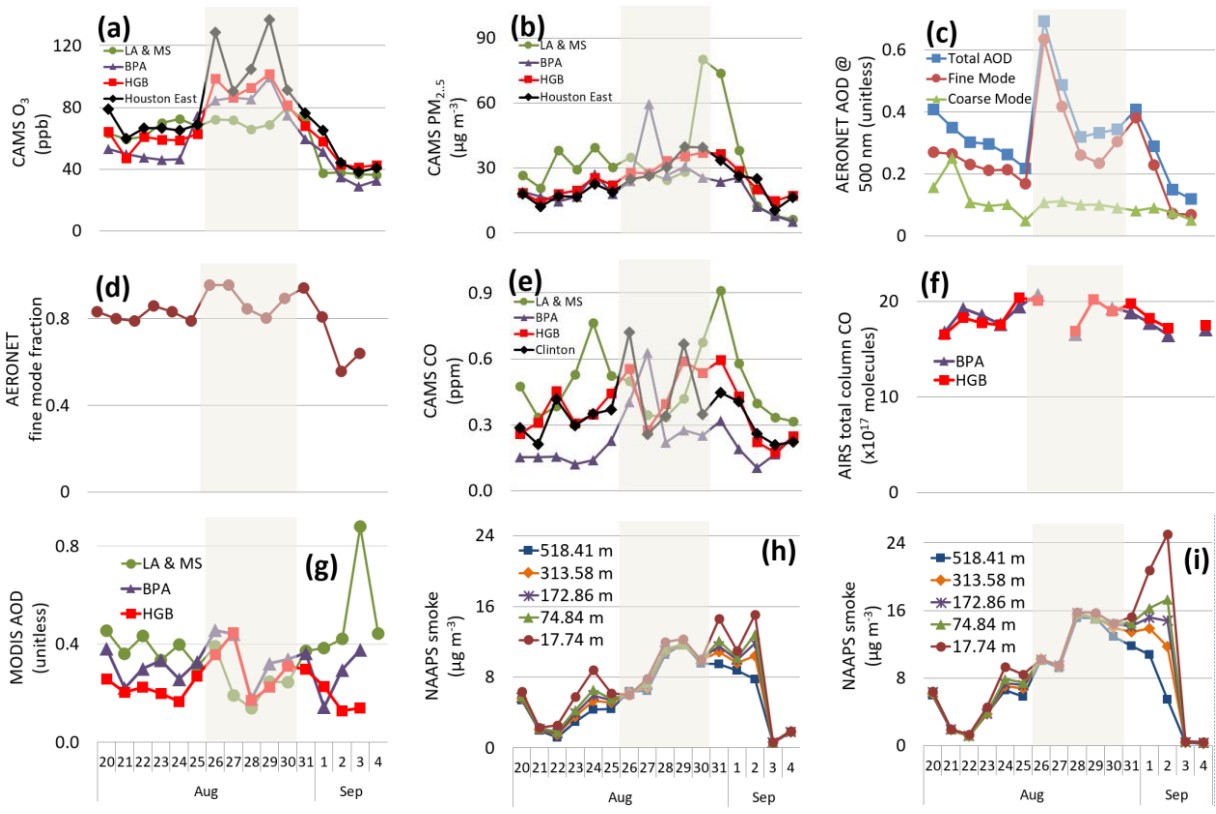

**Figure 2:** Time series plots for 20Aug-4Sep of: (a) means of daily maximum 1-hr O3 concentrations (ppb) at Houston East; (b) means of daily maximum 1-hr PM2.5 concentrations (µg m-3) at Houston East; (c) daily maximum total column aerosols by mode at 500 nm from Univ. of Houston AERONET site calculated from the Spectral Deconvolution Algorithm (SDA v4.1), level 1.5 cloud-screened without final calibrations, v2 direct sun algorithm; (d) daily maximum fine mode fraction AOD at 500 nm from Univ. of Houston AERONET site calculated as (c); (e) means of daily maximum 1-hr CO concentrations (ppm) at Clinton; (f) level 2 mean AIRS total column CO (×10$^{17}$ molecules) retrievals for pixels over HGB (29.0°N-30.5°N, 94.5°W-96.0°W) and BPA (29.5°N-30.25°N, 93.5°W-94.5°W); (g) daily mean





level 2 3-km resolution total column AOD (unitless) from MODIS Terra (MOD04_L2_3K_2011231 -- MOD04_L2_3K_2011243) and Aqua (MYD04_L2_3K_2011231 -- MYD04_L2_3K_2011243); (h) daily average NAAPS model smoke predictions (µg m$^{-3}$) for the 5 layers closest to the surface for the 4 1°× 1° grid cells over HGB (29.5°N-30.5°N, 94.5°W-95.5°W), 20Aug-4Sep; (i) daily average NAAPS model smoke predictions (µg m$^{-3}$) for the 5 layers closest to the surface for the 2 1°× 1° grid cells over BPA (29.5°N-30.5°N, 93.5°W), 20Aug-4Sep. AOD values above 1.5 are considered anomalous and were excluded. In plots of Houston East and Clinton, values are daily maxima rather than means. Sources: TCEQ Monitoring Division; datafed.net; aeronet.gsfc.nasa.gov; NASA.

**Figure 3:** Histograms for three periods (except where noted): 20-24Aug, 26-30Aug and 31Aug-04Sep for: daily maximum 1-hr O₃ concentrations (ppb) for 36 HGB monitors (a-1) and 9 BPA monitors (a-2); daily maximum 1-hr PM₂.₅ concentrations (µg m$^{-3}$) for 12 HGB monitors (b-1) and 4 BPA monitors (b-2); daytime hourly AERONET total column AOD at 500 nm (c-1), AERONET fine mode





AOD (c-2), and AERONET fine mode fraction AOD (c-3) from the Univ. of Houston site (Moody Tower); daily maximum 1-hr CO concentrations (ppb) for 4 HGB monitors (d-1) and 2 BPA monitors (d-2); level 2 mean AIRS total column CO ($\times 10^{17}$ molecules) retrievals for pixels over HGB (29.0°N-30.5°N, 94.5°W-96.0°W) (e-1) and BPA (29.5°N-30.25°N, 93.5°W-94.5°W) (e-2); and NAAPS model smoke predictions (µg m$^{-3}$) for the 5 layers closest to the surface for the 4 1°× 1° grid cells over HGB (29.5°N-30.5°N, 94.5°W-95.5°W) (f-1) and the 1°× 1° grid cell over BPA (29.5°N-30.5°N, 93.5°W) (f-2). AOD values above 1.5 are considered anomalous and were excluded. Sources: TCEQ Monitoring Division; datafed.net; aeronet.gsfc.nasa.gov; NASA.

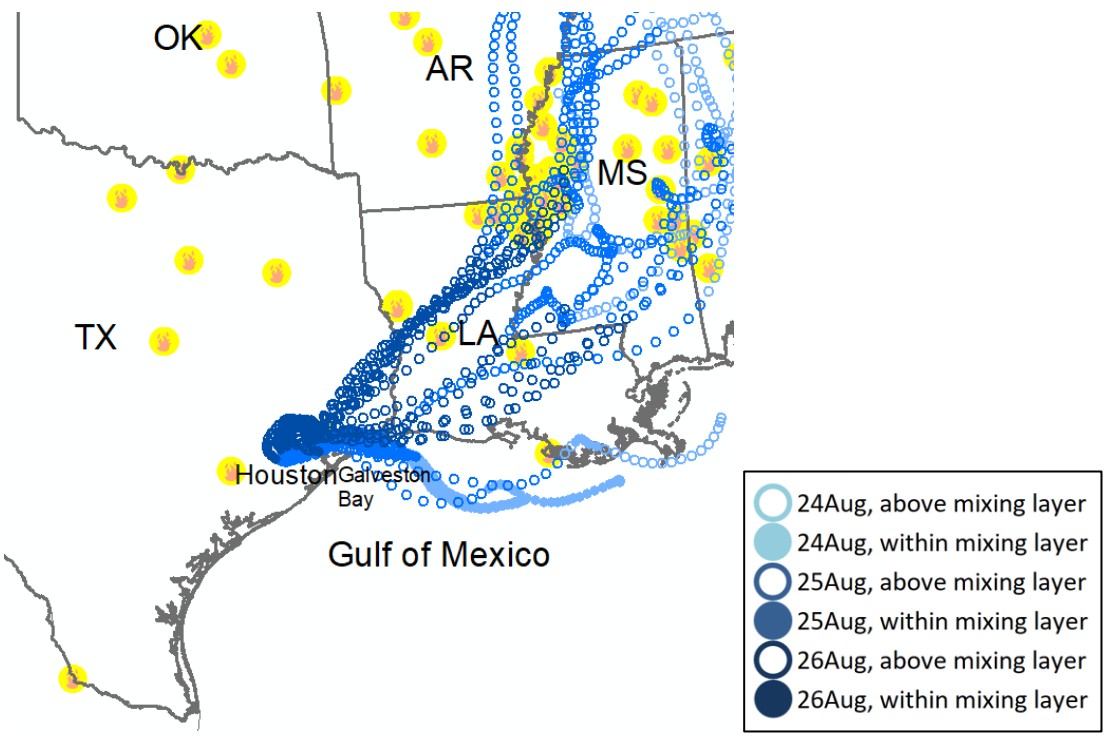

**Figure 4:** Active fires (flame icons) detected by MODIS, 24–26Aug, overlaid with 1-hr coordinates of HYSPLIT backward trajectories terminating at Houston East, 19:00 UTC on 26Aug at selected altitudes (10 m, 25 m, 50 m, 100 m, and at 250 m intervals from 250 m–5.0 km). Sources: modis-fire.umd.edu/pages/ActiveFire.php?target=Methodology; ready.arl.noaa.gov/HYSPLIT.php.

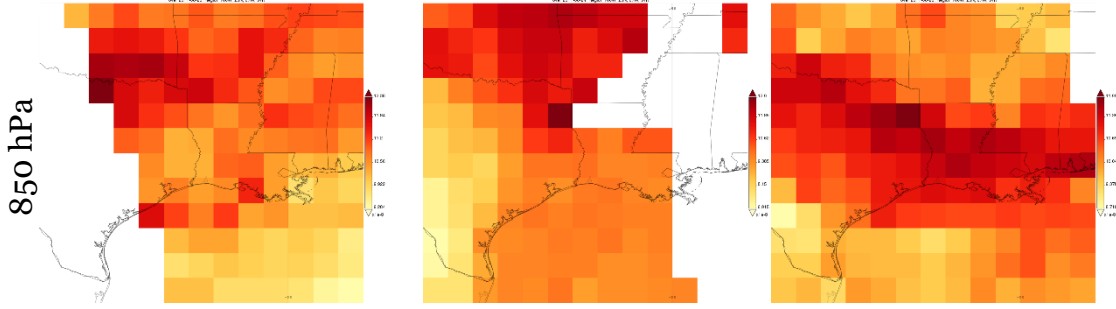



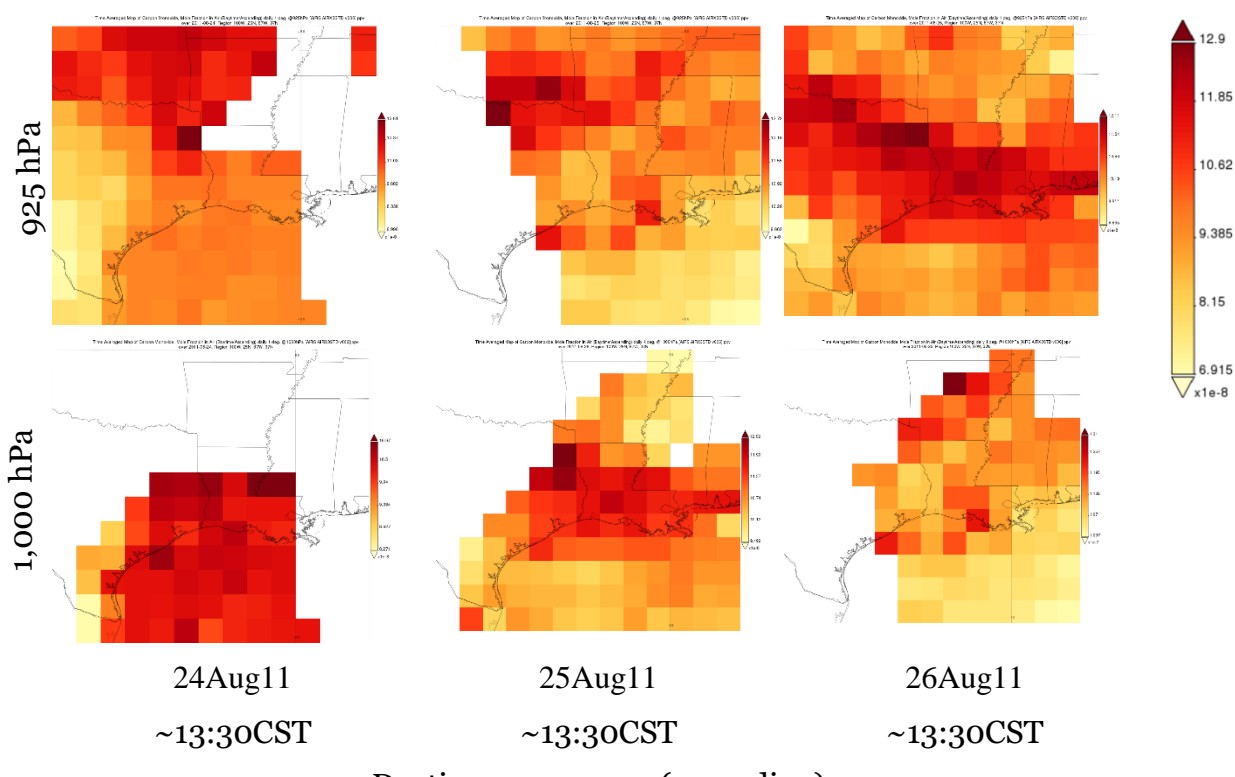

**Figure 5:** Time averaged maps of AIRS (Aqua) CO, mole fraction in air (daytime/ascending) daily 1°× 1°, August 24-26, 2011, at 3 atmospheric pressure levels: 850 hPa (top row), 925 hPa (center row) and 1,000 hPa (bottom row). Plots courtesy of Giovanni online data system, NASA GES DISC.

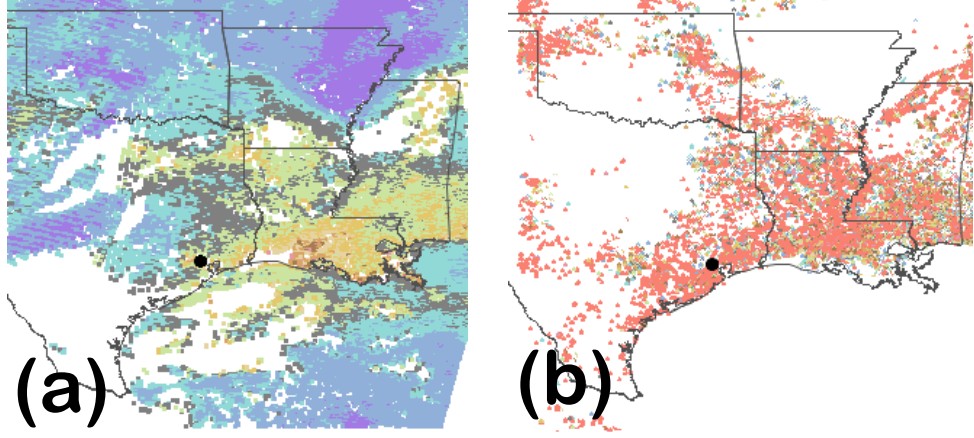



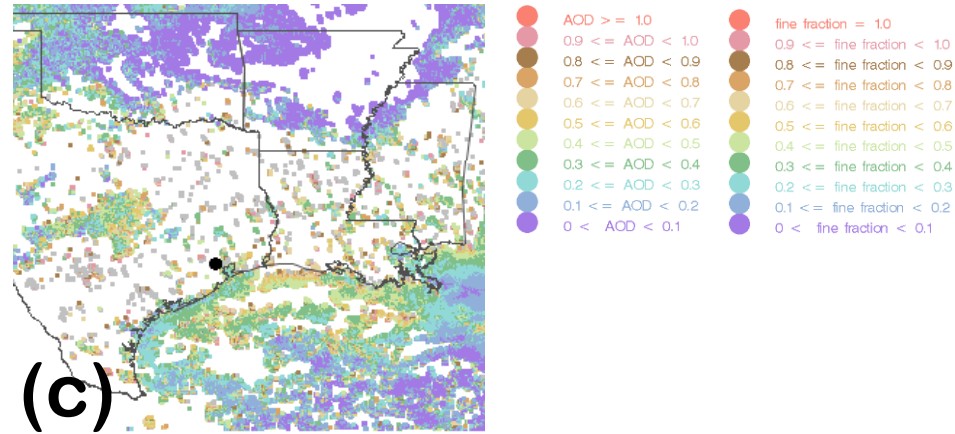

**Figure 6:** (a) Level 2 3-km resolution optical depth land and ocean (unitless) from MODIS Terra (MOD04_L2_3K_2011238), 26Aug; (b) level 2 3-km resolution optical depth ratio small land (fine mode fraction) from MODIS Terra (MOD04_L2_3K_2011238), 26Aug; (c) total column AOD retrieved by GOES13 ('GOES East') satellite over North America at 19:15 UTC, 26Aug (GOES13.GASP-AOD-GZ.J2011238.T0015Z). AOD values over 1.5 are considered anomalous and were excluded.

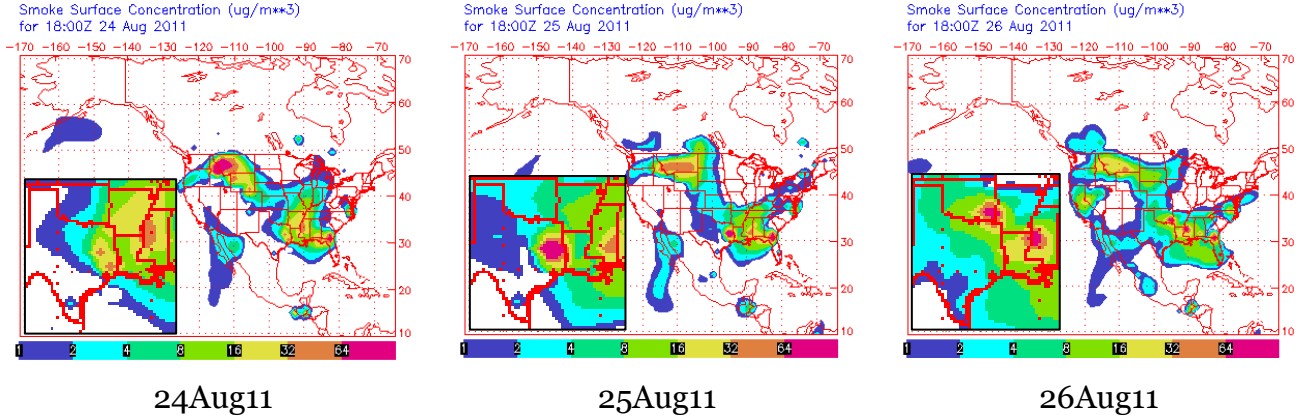

5    **Figure 7:** Estimates of surface smoke concentrations (µg m⁻³) from the NAAPS model at 18:00 UTC, 24-26Aug. Minimum contouring is 0.2 µg m⁻³. Insets are same time and date, zoomed to area of interest. www.nrlmry.navy.mil/aerosol_web/Docs/globaer_plots.html.





**Figure 8:** Level 2 CALIOP vertical feature mask aerosol subtype (CAL_LID_L2_VFM-ValStage1-V3-01.2011-08-26T19-14-29ZD) for 54 curtains in the vicinity of the HGB area, 26Aug2011 at ~19:38Z. Left: CALIPSO flight track. Crosses are locations of 5-km curtains with transition points marked with larger labelled crosses (a, b, c, d). Right: CALIOP vertical feature mask for lowest 8 km of each 5-km curtain with transition points labelled (a, b, c, d).

