# Peer review of "Identifying biomass burning impacts on air quality in Southeast Texas 26–29 August 2011 using satellites, models and surface data"

_Atmospheric Chemistry and Physics, 2017_

## Referee Comment (RC1) · Anonymous Referee #1 · 21 Jan 2018

This manuscript presents a wide array of evidence to support the claim that biomass burning fires in Louisiana and Mississippi contributed to a several-day air pollution event in southeast Texas in August 2011. The array of evidence includes ground-level observations of ozone, aerosols, and CO; meteorological back trajectories; satellite observations of fires and air pollutants; and an atmospheric model.

As its main objective, the article asserts that it "demonstrates an approach to identifying biomass burning influences on high ozone events which may be useful in determining compliance with EPA NAAQS. ... This approach could well be adapted for application

to other pollution events in the HGB area as well as in other regions and at other times." This claim is true in the sense that the methods could be adapted and applied readily by a regulatory agency. The back trajectories are straightforward to run, and almost all of the observations and modeling are publicly available and conducted by others. Whether it would be at all useful for NAAQS compliance is more questionable; EPA tends to be skeptical of claims of "exceptional events", and this method cannot quantify the amount of ozone contributed by the out-of-state fires. The authors do not even demonstrate that the event is exceptional, despite their claim that "this evidence clearly demonstrates that O3 events on 26Aug and 29Aug were unusual even for this area." No evidence is presented of a long-term record to show whether these events were exceptional or even unusual.

In terms of scientific merit, the contributions of this paper are thin. No new methods or rigorous analysis are introduced, so the main value is in demonstrating the wide array of available data and modeling results that can be assembled readily. Nevertheless, the methods are sound and the presentation is clear. Thus, the revisions described below are relatively minor in order for the paper to be publishable in some journal. The main judgment for the Editor to make is whether this paper's compilation of outside data and modeling results to analyze a single episode in a single region is sufficient to merit publication in ACP.

Specific comments: 1. 2011 was a year of extreme drought and wildfires for Texas. This should be noted, though the authors find evidence that the smoke for this particular episode was from out of state. 2. Nothing is done to show how exceptional or unusual this event was. Thus, the paper does not "demonstrate that O3 events on 26Aug and 29Aug were unusual." (p. 3, line 14) 3. In several instances, such as p. 3, lines 21-22, and p. 5, lines 6-8, the authors give the impression that the meteorology of the Houston region makes it prone to high ozone. In fact, despite its occasional episodes of high ozone, on an average summer day Houston has less ozone than most cities and even some rural areas, thanks to its favorable meteorology of inflow from the Gulf

of Mexico. That inflow also helps keep PM levels within EPA limits, despite the region's large population, heavy traffic, and numerous industrial sources. And stratosphere-troposphere exchange ozone events (p. 5, line 9) are infrequent in this region relative to mountainous regions. Yes, there are meteorological conditions such as the post-frontal conditions of this episode or other stagnant periods that are conducive to the high ozone that leads the region to non-attainment. But the article perpetuates mistaken impressions about the frequency of polluted days in Houston and the favorability of its meteorology for ozone formation beyond occasional episodes. 4. The abstract and conclusions mention the use of the CMAQ model, but that is not presented in the paper. 5. The ozone data plotted in Figure 1 are inconsistent with a claim that out-of-state biomass burning played a dominant role during this episode. Peak ozone concentrations vary by a factor of 2 across the Houston region, reflecting a typical pattern of a sharp gradient between ozone upwind and downwind of the region's main local emissions sources. Pollution traveling from days-away fires would have a more spatially uniform pattern. It is less clear from the PM/AOD/CO data whether there is a broad-based contribution from out-of-state fires. 6. I find it difficult to glean much of value from the numerous histograms in Figure 3.

---

## Referee Comment (RC2) · Anonymous Referee #2 · 13 Mar 2018

This paper attempts to show the impacts from biomass burning on SE Texas (Houston area) from multiple biomass burning events in August 2011. The stated goals are to demonstrate an impact on surface O3 from the bb emissions. The analysis is a hodge-podge of surface observations, models and satellite data that tries to show the link with surface O3. Unfortunately none of these really convincingly link the bb emissions to O3. One can find many bb events, satellite data and even trajectories that purport to show a link, but often the actual concentrations are very low. How can we say that high O3 in Houston (a very high O3 city) was due to bb emissions? What are the concrete pieces of evidence that support transport of smoke into the city and how much was O3 enhanced by this process? So this analysis (and manuscript) needs a major redo before it can demonstrate something useful. To guide this, I suggest the authors consider, at minimum, these questions: 1. What is the proof that PM, O3 or its precursors ( CO, VOCs and/or NOx) were transported into Houston at that time? 2. Are there specific tracers that could be used to identify smoke influence at the surface (e.g. enhancement ratios, pattern of VOCs, potassium or other bb tracers, etc). 3. Does high PM prove that smoke was transported? 4. Were PM and O3 correlated on these days or does this matter? 5. Why do the observations show a wide range in highest days (eg highest O3 on 8/26 and 8/29, highest PM on 8/30 and 31, highest AOD on 8/26, highest NAAPS on 9/2). 6. If O3 was enhanced by the bb emissions, by how much and why isn't O3 enhanced on days with highest PM? Are there other factors (e.g. temp, meteorology, etc) that are needed to explain this? A few other comments: Abstract: The abstract states "...we examine the influence of transported emissions ....on O3 and precurrsors..." But most of the analysis is focused on the satellite data and models. If the goal is to demonstrate surface impacts, the authors need to spend more time analyzing and presenting the surface data. Most of the surface data presentation uses daily means, which is insufficient to understand what is going on. While the introduction and background section include a lot of citations, most are 2010 or earlier. The authors need to update these citations to include more recent findings on O3 and biomass burning influence.

Figure 2: These demonstrate that peaks occur on random days throughout the period. It is not clear what is the connection between any of these. And none of this "proves" the presence of smoke. Figure 3: Very hard to decipher. Caption says histograms, but this figure does not show a usual histograms and the legends are hard to read (fonts too small). What are you trying to show here? Does this figure show something that is not in figure 2?

Figure 4: I think a key missing point is that fires are very often present in the Mississippi

[Figure]

Valley. The fact that trajectories go by fires in no way proves that these fires had a significant impact. You need a stronger case to make that claim. Is PM much higher than usual for this trajectory direction on 8/26 and 8/29? Figure 7: All models seem to have a hard time getting bb transport right and NAAPS is no exception. It's a challenging problem for many reasons. I note from Figure 2, that NAAPS predicts highest PM on 9/2, whereas in reality it occurred on 8/30. So what do we take away from this?

—————————————————————

---

## Author Comment (AC1) · 21 May 2018

Westenbarger and Gary A. Morris

Anonymous Referee #1

This manuscript presents a wide array of evidence to support the claim that biomass burning fires in Louisiana and Mississippi contributed to a several-day air pollution

event in southeast Texas in August 2011. The array of evidence includes ground-level observations of ozone, aerosols, and CO; meteorological back trajectories; satellite observations of fires and air pollutants; and an atmospheric model. As its main objective, the article asserts that it "demonstrates an approach to identifying biomass burning influences on high ozone events which may be useful in determining compliance with EPA NAAQS. ... This approach could well be adapted for application to other pollution events in the HGB area as well as in other regions and at other times." This claim is true in the sense that the methods could be adapted and applied readily by a regulatory agency. The back trajectories are straightforward to run, and almost all of the observations and modeling are publicly available and conducted by others. Whether it would be at all useful for NAAQS compliance is more questionable; EPA tends to be skeptical of claims of "exceptional events", and this method cannot quantify the amount of ozone contributed by the out-of-state fires.

Author response: While we recognize that it would be very useful to present model results, this paper presents an observational approach that compares surface O3 concentrations from prior days to one of the days in question (26Aug11) to calculate an approximate O3 enhancement of 63-71 ppb. A second approach, not in the original version but integrated into the revised manuscript, is provided by examining data from an ozonesonde flight from the University of Houston on 29Aug11, the second day of concern during the period. From Figure R-1.1 below, we identify a surface O3 enhancement of about 76 ppb relative to the lower free troposphere (subtract the O3 mixing ratio of 136 ppb in the boundary layer from the O3 mixing ratio of 60 ppb in the lower free troposphere at about 3.5 km altitude). This difference represents a surface enhancement substantially in excess of a typical bad air day in the HGB region, as supported by Figure R-1.2 (below) which plots these computed gradients between the boundary layer and lower free troposphere for all 600+ soundings in the Houston region from 2004 to 2016. This plot shows that the enhancement observed on 29Aug11 (day 241) is quite exceptional over a considerable historical record (> 1 decade), as it would not even appear on this chart, which ranges from -40 to +40 ppb enhancements/disenhancements. This second approach strengthens the prior result as the computed value (76 ppb) is substantially similar to the first (63-71 ppb), which appears in the original version of the paper. Also, a new paragraph [32] has been added for clarification:

"A final piece of the puzzle is provided in Figure 9, which plots O3, relative humidity and potential temperature ('theta') measurements taken by an ozonesonde instrument on a flight conducted on 29Aug from the campus of the University of Houston. In the plot, the extreme gradient in O3 between the boundary layer at roughly 2.5 km altitude (∼136 ppb) and the lower free troposphere at roughly 3.5 km (∼60 ppb), is clear. This difference of ∼76 ppb O3 is surprisingly similar to the result obtained using surface monitors: 63–71 ppb, and is evidence of the extreme nature of the O3 regime impacting the HGB area during this event."

Figure R-1.1 Source: http://physics.valpo.edu/ozone/texasdata/20110829/trop_o3mr_2011082921.pdf

Figure R-1.2

The authors do not even demonstrate that the event is exceptional, despite their claim that "this evidence clearly demonstrates that O3 events on 26Aug and 29Aug were unusual even for this area." No evidence is presented of a long-term record to show whether these events were exceptional or even unusual. Author response: Please see "August 26, 2011 Exceptional Events Demonstration Package for the Houston-Galveston-Brazoria One-hour Ozone Nonattainment Area" submitted to U.S. EPA by the Texas Commission on Environmental Quality (https://www.tceq.texas.gov/assets/public/implementation/air/sip/hgb/Exceptional_Event_TSD_Aug_26_2011_Public_Com specifically Figure 3-1 on page 3-2 (replicated as Figure R-1.3 below) which shows that daily maximum O3 on 26Aug11 was well above the 99th percentile for the season (late summer) in recent years (2009-2011). We incorporate this reference in the revised version of the paper.

Figure R-1.3

[Figure]

In terms of scientific merit, the contributions of this paper are thin. No new methods or rigorous analysis are introduced, so the main value is in demonstrating the wide array of available data and modeling results that can be assembled readily. Nevertheless, the methods are sound and the presentation is clear. Thus, the revisions described below are relatively minor in order for the paper to be publishable in some journal. The main judgment for the Editor to make is whether this paper's compilation of outside data and modeling results to analyze a single episode in a single region is sufficient to merit publication in ACP.

Specific comments:

1. 2011 was a year of extreme drought and wildfires for Texas. This should be noted, though the authors find evidence that the smoke for this particular episode was from out of state.

Author response: Text has been inserted in paragraph [6] to indicate that 2011 was a very dry year and experienced a larger than normal number of wild fires. The new text: "The year 2011 was noteworthy in North America for extreme drought in the southern Plains region of the U.S. and high incidence of wild fires, recording the 3rd most acres burned over the prior 12 years (2000-2011) (NOAA, 2012). In 2014, the Texas Commission on Environmental Quality (TCEQ) submitted to the U.S. EPA a document (TCEQ, 2014) demonstrating that the daily maximum 1-hr O3 concentration measured at the Houston East monitor on 26Aug2011 was substantially above the 99th percentile for that time of year when compared to recent years, 2009-2011 (see Figure 3-1 on page 3-2 of that document)."

2. Nothing is done to show how exceptional or unusual this event was. Thus, the paper does not "demonstrate that O3 events on 26Aug and 29Aug were unusual." (p. 3, line 14)

Author response: Please see response above.

3. In several instances, such as p. 3, lines 21- 22, and p. 5, lines 6-8, the authors give the impression that the meteorology of the Houston region makes it prone to high ozone. In fact, despite its occasional episodes of high ozone, on an average summer day Houston has less ozone than most cities and even some rural areas, thanks to its favorable meteorology of inflow from the Gulf of Mexico. That inflow also helps keep PM levels within EPA limits, despite the region's large population, heavy traffic, and numerous industrial sources. And stratosphere-troposphere exchange ozone events (p. 5, line 9) are infrequent in this region relative to mountainous regions. Yes, there are meteorological conditions such as the post-frontal conditions of this episode or other stagnant periods that are conducive to the high ozone that leads the region to non-attainment. But the article perpetuates mistaken impressions about the frequency of polluted days in Houston and the favorability of its meteorology for ozone formation beyond occasional episodes.

Author response: Due in large part to unique coastal meteorology, the HGB area is prone to experiencing two periods of high O3 during the year: one in spring and one in late summer, with lower peaks in early- and mid-summer. A number of studies have examined the various aspects of local, regional, and continental-scale meteorology that impact the area. Text has been inserted in paragraph [11] to describe the seasonality of the meteorological influence in the HGB area and list several studies on this topic. The new text:

"A substantial body of evidence has demonstrated that this coastal meteorology coupled with larger-scale influences from the Gulf of Mexico and Atlantic Ocean, such as seasonal westward migration of the Bermuda High pressure system, and the location of the city downwind of many emissions sources, subjects the HGB area to particularly O3-conducive conditions during spring and late summer, though less so in mid-summer (see e.g., Nielsen-Gammon, 2005a; Nielsen-Gammon, 2005b; Wang, 2016; Wang, 2015; Berlin, 2013; Darby, 2005)."

4. The abstract and conclusions mention the use of the CMAQ model, but that is not

presented in the paper.

Author response: Discussion of the CMAQ model has been removed from the abstract.

5. The ozone data plotted in Figure 1 are inconsistent with a claim that out-of-state biomass burning played a dominant role during this episode. Peak ozone concentrations vary by a factor of 2 across the Houston region, reflecting a typical pattern of a sharp gradient between ozone upwind and downwind of the region's main local emissions sources. Pollution traveling from days-away fires would have a more spatially uniform pattern. It is less clear from the PM/AOD/CO data whether there is a broad-based contribution from out-of-state fires.

Author response: Local sources always contribute to O3 gradients in the HGB area, even in the absence of transported pollution, and these spatial gradients can be quite substantial. Transport enhances the entire distribution in a more or less uniform manner, as the commenter notes. This paper demonstrates that the combination of regional-scale long-range transport-related O3 enhancements super-imposed over typical local O3 variability led to the exceptional O3 concentrations measured at surface monitors.

6. I find it difficult to glean much of value from the numerous histograms in Figure 3.

Author response: The main point of Figure 3 is to present a recurring pattern of vastly changing region-wide distributions across 3 distinct time periods for multiple parameters. In each case, the middle period, which is most strongly influenced by transport of ozone and precursors from biomass burning emissions, appears substantially different compared to the periods before and after. Comparisons of histograms across the 'before', 'during' and 'after' time periods, replicated across numerous parameters, are critical to demonstrating the pervasive and compelling influence of biomass burning emissions on O3 in the HGB area exhibited in the 'during' period. We have revised the figure and text in hopes of clarifying its contents (see paragraph [9]).

[Figure]

Please also note the supplement to this comment:
https://www.atmos-chem-phys-discuss.net/acp-2017-1234/acp-2017-1234-AC1-supplement.pdf
* * *
* * *
**Houston, TX - 2011082921**

O₃ (a)  (d)

RH (a)  (d)

Theta (a) (d)

**Fig. 1.** Figure R-1.1 Ozone, rh and potential temperature (theta) for flight from University of Houston campus, 29Aug11

[Figure]

**Figure 3-36.** The difference between the mean ozone in the free troposphere and the boundary layer as found in ozonesonde profiles over Houston and as a function of day of the year. The color coding of the dots is associated with the three cases described in the text.

**Fig. 2.** Figure R-1.2 Houston seasonal vertical O3 gradient by day of year

Ozone (ppb)

08/26/11

99th Percentile

Percentile

**Figure 3-1: Percentile Rankings of Seasonal Daily Maximum One-Hour Ozone Concentrations at the Houston East (CAMS 1) Monitoring Site from 2009 through 2011**

**Fig. 3.** Figure R-1.3 Percentile rankings of seasonal daily maximum one-hour ozone concentrations at the Houston East (CAMS 1) monitoring station, 2009-2011

---

## Author Comment (AC2) · 21 May 2018

Westenbarger and Gary A. Morris

Anonymous Referee #2

This paper attempts to show the impacts from biomass burning on SE Texas (Houston area) from multiple biomass burning events in August 2011. The stated goals are to

demonstrate an impact on surface O3 from the bb emissions. The analysis is a hodge-podge of surface observations, models and satellite data that tries to show the link with surface O3. Unfortunately none of these really convincingly link the bb emissions to O3. One can find many bb events, satellite data and even trajectories that purport to show a link, but often the actual concentrations are very low. How can we say that high O3 in Houston (a very high O3 city) was due to bb emissions?

Author response: We have compiled multiple data sources that we believe have made a compelling case, adding ozonesonde data in the revised version. See response to Reviewer #1.

What are the concrete pieces of evidence that support transport of smoke into the city and how much was O3 enhanced by this process?

Author response: Satellite instruments, surface measurements and ozonesonde profiles. In the original manuscript, we provided one estimate of the enhancement by comparing surface O3 concentrations from prior days to one of the days in question (26Aug11) to calculate an approximate O3 enhancement of 63-71 ppb. A second approach, not in the original version but integrated into the revised manuscript, is provided by examining data from an ozonesonde flight from the University of Houston on 29Aug11, the second day of concern during the period. From Figure R-1.1 below, we identify a surface O3 enhancement of about 76 ppb relative to the lower free troposphere (subtract the O3 mixing ratio of 136 ppb in the boundary layer from the O3 mixing ratio of 60 ppb in the lower free troposphere at about 3.5 km altitude). This difference represents a surface enhancement substantially in excess of a typical bad air day in the HGB region, as supported by Figure R-1.2 below which plots these computed gradients between the boundary layer and lower free troposphere for all 600+ soundings in the Houston region from 2004 to 2016. This plot shows that the enhancement observed on 29Aug11 is quite exceptional over a considerable historical record (> 1 decade), as it would not even appear on this chart, which ranges from -40 to +40 ppb enhancements/disenhancements. This second approach strengthens the prior result

as the computed value (76 ppb) is substantially similar to the first (63-71 ppb), which appears in the original version of the paper.

So this analysis (and manuscript) needs a major redo before it can demonstrate something useful. To guide this, I suggest the authors consider, at minimum, these questions:

1. What is the proof that PM, O3 or its precursors (CO, VOCs and/or NOx) were transported into Houston at that time?

Author response: The authors believe the evidence from satellite instruments and backward trajectories are well-described in the paper. This evidence includes retrievals of smoke and aerosols from several satellite instruments as well as two independent sources of backward trajectories showing transport from areas with many active fires. We also put surface monitor data in context to show that during the event, the HGB environment appears different than typical.

2. Are there specific tracers that could be used to identify smoke influence at the surface (e.g. enhancement ratios, pattern of VOCs, potassium or other bb tracers, etc). Author response: Unfortunately, many of the best tracers (e.g., PAN, levoglucosan) are not available, since these generally are only measured during targeted studies using special equipment, not retrospective analyses such as this one. However, this paper does present indicators such as the ratio between fine and coarse particulate matter at the AERONET surface station and the aerosol subtype from the CALIOP instrument aboard CALIPSO. These measures support the conclusion that smoke from biomass burning was present. Other measures build upon these to suggest that this smoke contributed to O3 enhancements.

3. Does high PM prove that smoke was transported?

Author response: By itself, the presence of PM does not prove that smoke was transported. However, fine mode fraction data and the wide geographic range of affected

monitors are suggestive of biomass burning influences.

4. Were PM and O3 correlated on these days or does this matter?

Author response: Typically, O3 and PM are not correlated. However, on the days in question, they were. In the HGB area, PM generally increases as the boundary layer shrinks while O3 decreases. Figure 3 demonstrates that PM and O3 increased simultaneously during the period in question. This figure also shows that PM decreased at the end of the period, though not as precipitously as O3, and that this was likely due to the change in meteorological regimes and the O3 production rate due to enhanced clouds with arrival of a cold front. Finally, Figure 3(b-1) shows that all 12 PM monitors measured PM2.5 exceeding 20 $\mu$g m-3 in the middle period, a substantial enhancement over the "before" period.

5. Why do the observations show a wide range in highest days (eg highest O3 on 8/26 and 8/29, highest PM on 8/30 and 31, highest AOD on 8/26, highest NAAPS on 9/2).

Author response: O3 photochemistry is very complex and involves many factors, including precursors, meteorology, and sunlight. Presence of precursors alone is not sufficient to lead to high O3. Also, 9/2 was strongly influenced by Tropical Storm Lee to the east of Houston.

6. If O3 was enhanced by the bb emissions, by how much and why isn't O3 enhanced on days with highest PM? Are there other factors (e.g. temp, meteorology, etc) that are needed to explain this? A few other comments: Abstract: The abstract states ". . .we examine the influence of transported emissions....on O3 and precursors. . ." But most of the analysis is focused on the satellite data and models. If the goal is to demonstrate surface impacts, the authors need to spend more time analyzing and presenting the surface data. Most of the surface data presentation uses daily means, which is insufficient to understand what is going on.

Author response: Technically, the analyses in the paper use means of daily maxima,

not means alone. The authors believe examination of these observations from the high ends of the distributions are more informative. As for the amount of the O3 enhancement, we referenced 63-71 ppb in the Abstract and 70+ ppb from the new ozonesonde data.

While the introduction and background section include a lot of citations, most are 2010 or earlier. The authors need to update these citations to include more recent findings on O3 and biomass burning influence.

Author response: The references have been updated to include more recent research.

Figure 2: These demonstrate that peaks occur on random days throughout the period. It is not clear what is the connection between any of these. And none of this "proves" the presence of smoke. Figure 3: Very hard to decipher. Caption says histograms, but this figure does not show a usual histograms and the legends are hard to read (fonts too small). What are you trying to show here?

Author response: It is important to note that several time series plots in Figure 2 include multiple regions. Peaks do not align temporally because of the nature of transport timing, meteorology and recirculation influences, with LA/MS impacted first, followed by BPA then HGB. These may appear random because of the overlapping regional plots. Figure 3 has been revised to improve legibility and increase its size. Text has been revised to clearly note that each panel contains six histograms (except for Figure 3(c-3) which includes only 3) (See paragraph [9]).

Does this figure show something that is not in figure 2?

Author response: Yes. Both the time series and histograms provide important insights. However, we have revised both for clarity. The histograms demonstrate well the changing pollution regimes from before to during to after the event.

Figure 4: I think a key missing point is that fires are very often present in the Mississippi Valley. The fact that trajectories go by fires in no way proves that these fires had a

significant impact. You need a stronger case to make that claim. Is PM much higher than usual for this trajectory direction on 8/26 and 8/29?

Author response: Wind directions were highly variable on 8/26, less so on 8/29. On 8/26, winds rotated clockwise around the city before arriving at the target surface monitoring station, as backward trajectories show. So, identifying a source direction from available wind direction data is complicated. However, the below Figure R-2.1 shows wind roses for all stations in the metro area, both frequencies (left; grey rings are a frequency of 30) and peak PM2.5 concentrations (right; grey rings are 60 $\mu$g/m-3) by wind direction (binned into 16 directional bins) on each of the days during the study period. On both 8/26 and 8/29, winds arrived from many directions so there are more, shorter barbs, than on other days (left). Much higher PM2.5 concentrations are evident on the "during" days (26-30Aug) than the period "before", especially arriving from the direction of the Gulf of Mexico (generally, southeast, south and southwest). However, as noted, wind variability makes even these source directions questionable. This may be more easily detected in the tables presented in Figure R-2.2 which show frequencies (left) and daily peak PM2.5 concentrations (right) in the HGB area by day and 16 wind direction bins. In the "before" period, except for a single northerly signal, most winds arrived from southerly or nearly southerly directions and the PM2.5 contained in those parcels was relatively low. In the "during" period, winds arrived from more directions and PM2.5 concentrations were generally higher from all directions. Finally, in the "after" period, variable winds at the beginning (31Aug) became dominated by Tropical Storm Lee to the east of the HGB area, as winds shifted to almost entirely northerly or near-northerly, and PM2.5 concentrations dropped considerably.

Figure R-2.1a

Figure R-2.1b

Figure R-2.2

Figure R-2.2b

Figure 7: All models seem to have a hard time getting bb transport right and NAAPS is no exception. It's a challenging problem for many reasons. I note from Figure 2, that NAAPS predicts highest PM on 9/2, whereas in reality it occurred on 8/30. So what do we take away from this?

Author response: Models often do not match observations for a variety of reasons. Our intention in this paper was to focus on observations, while using model predictions to supplement these. The authors believe that the observations alone provide a compelling case.

Please also note the supplement to this comment:
https://www.atmos-chem-phys-discuss.net/acp-2017-1234/acp-2017-1234-AC2-supplement.pdf
* * *
**Frequencies of (resultant) wind direction at HGB monitors**

| 20-Aug | 21-Aug | 22-Aug | 23-Aug | 24-Aug |
|---|---|---|---|---|

| 26-Aug | 27-Aug | 28-Aug | 29-Aug | 30-Aug |
|---|---|---|---|---|

| 31-Aug | 1-Sep | 2-Sep | 3-Sep | 4-Sep |
|---|---|---|---|---|

**Fig. 1.** Figure R-2.1a. Frequencies of (resultant) 16 wind direction bins at HGB monitors, 20Aug-04Sep11

**Maximum PM2.5 concentrations by wind direction at HGB monitors**

| 20-Aug | 21-Aug | 22-Aug | 23-Aug | 24-Aug |
|---|---|---|---|---|

| 26-Aug | 27-Aug | 28-Aug | 29-Aug | 30-Aug |
|---|---|---|---|---|

| 31-Aug | 1-Sep | 2-Sep | 3-Sep | 4-Sep |
|---|---|---|---|---|

**Fig. 2.** Figure R-2.1b. Maximum PM2.5 concentrations by 16 wind direction bins at HGB monitors, 20Aug-04Sep11

| | Aug | | | | | | Aug | | | | | | Aug | Sep | | | |
|---|---|---|---|---|---|---|---|---|---|---|---|---|---|---|---|---|---|
| | 20 | 21 | 22 | 23 | 24 | | 26 | 27 | 28 | 29 | 30 | | 31 | 1 | 2 | 3 | 4 |
| N | 12 | 12 | 15 | 12 | 13 | | 20 | 15 | 14 | 14 | 14 | | 13 | 16 | 12 | 51 | 45 |
| NNE | | | 2 | | 3 | | 2 | 4 | 2 | | 1 | | 2 | 10 | 12 | 20 | 12 |
| NE | | 1 | 3 | | 1 | | | 1 | 7 | 1 | 1 | | 4 | 10 | 28 | 1 | |
| ENE | | 2 | 1 | | 4 | | | 2 | 7 | 2 | 3 | | 3 | 10 | 28 | | |
| E | | 2 | | 0 | 2 | | 1 | 1 | 5 | 3 | 3 | | 5 | 26 | 16 | | |
| ESE | 7 | 6 | 5 | 4 | 12 | | 3 | 2 | 9 | 5 | 13 | | 16 | 14 | 4 | | |
| SE | 13 | 14 | 14 | 9 | 15 | | 5 | | 9 | 11 | 26 | | 26 | 12 | 1 | | |
| SSE | 18 | 15 | 7 | 10 | 6 | | 10 | 2 | 7 | 10 | 22 | | 15 | | | | |
| S | 15 | 21 | 20 | 14 | 8 | | 11 | 5 | 6 | 10 | 9 | | 6 | 1 | | | |
| SSW | 14 | 14 | 13 | 11 | 13 | | 9 | 10 | 9 | 8 | 1 | | 5 | | | | |
| SW | 9 | 7 | 8 | 15 | 11 | | 9 | 11 | 14 | 11 | 2 | | 1 | | | | |
| WSW | 6 | 5 | 8 | 15 | 6 | | 6 | 16 | 6 | 9 | 2 | | 1 | | | | |
| W | 4 | 2 | 4 | 7 | 5 | | 2 | 18 | 4 | 6 | 2 | | 1 | 1 | | | |
| WNW | 0 | | 1 | 1 | 2 | | 6 | 10 | 2 | 4 | 1 | | | | | | |
| NW | 1 | 1 | 1 | 0 | 1 | | 9 | 3 | 1 | 1 | 1 | | 1 | 1 | | 3 | 7 |
| NNW | | | 2 | | 1 | | 8 | 2 | 1 | 3 | | | 2 | 1 | | 25 | 36 |

**Fig. 3.** Figure R-2.2a. Frequencies of (resultant) 16 wind direction bins at HGB monitors, 20Aug-04Sep11

**Aug**

| | 20 | 21 | 22 | 23 | 24 |
|---|---|---|---|---|---|
| N | 20 | 13 | 21 | 18 | 13 |
| NNE | | | 12 | | 17 |
| NE | | | 11 | 23 | 11 |
| ENE | | | 13 | 16 | 88 |
| E | | 16 | | 20 | 18 |
| ESE | 20 | 15 | 17 | 33 | 32 |
| SE | 18 | 20 | 17 | 46 | 46 |
| SSE | 19 | 15 | 22 | 30 | 34 |
| S | 17 | 13 | 20 | 20 | 11 |
| SSW | 21 | 22 | 13 | 16 | 26 |
| SW | 18 | 10 | 15 | 17 | 16 |
| WSW | 18 | 14 | 13 | 16 | 19 |
| W | 16 | 6 | 18 | 16 | 19 |
| WNW | 13 | | 15 | 11 | 13 |
| NW | 9 | 8 | 10 | 12 | 14 |
| NNW | | | 18 | | 7 |

**Aug**

| | 26 | 27 | 28 | 29 | 30 |
|---|---|---|---|---|---|
| N | 30 | 29 | 37 | 40 | 34 |
| NNE | 20 | 18 | 20 | | 24 |
| NE | | 14 | 24 | 21 | 22 |
| ENE | | 25 | 36 | 26 | 35 |
| E | 17 | 28 | 28 | 47 | 46 |
| ESE | 42 | 24 | 53 | 65 | 40 |
| SE | 28 | | 41 | 41 | 57 |
| SSE | 47 | 39 | 40 | 31 | 42 |
| S | 36 | 25 | 36 | 32 | 38 |
| SSW | 26 | 24 | 31 | 25 | 24 |
| SW | 25 | 24 | 29 | 28 | 23 |
| WSW | 23 | 26 | 22 | 36 | 24 |
| W | 16 | 24 | 19 | 37 | 21 |
| WNW | 17 | 19 | 13 | 39 | 17 |
| NW | 25 | 15 | 5 | 19 | 17 |
| NNW | 27 | 22 | 9 | 28 | |

**Aug  Sep**

| | 31 | 1 | 2 | 3 | 4 |
|---|---|---|---|---|---|
| N | 32 | 31 | 11 | 29 | 22 |
| NNE | 30 | 33 | 10 | 29 | 16 |
| NE | 53 | 34 | 30 | 0 | |
| ENE | 45 | 21 | 19 | | |
| E | 38 | 33 | 24 | | |
| ESE | 63 | 41 | 25 | | |
| SE | 49 | 20 | 15 | | |
| SSE | 28 | | | | |
| S | 27 | 15 | | | |
| SSW | 28 | | | | |
| SW | 21 | | | | |
| WSW | 29 | | | | |
| W | 27 | 14 | | | |
| WNW | | | | | |
| NW | 22 | 13 | | 5 | 8 |
| NNW | 31 | 15 | | 12 | 13 |

**Fig. 4.** Figure R 2.2b. Maximum PM2.5 concentrations by 16 wind direction bins at HGB monitors, 20Aug-04Sep11